# The Role of NAD^+^ in Metabolic Regulation of Adipose Tissue: Implications for Obesity-Induced Insulin Resistance

**DOI:** 10.3390/biomedicines11092560

**Published:** 2023-09-18

**Authors:** Tatjana Ruskovska, David A. Bernlohr

**Affiliations:** 1Faculty of Medical Sciences, Goce Delcev University, 2000 Stip, North Macedonia; tatjana.ruskovska@ugd.edu.mk; 2Department of Biochemistry, Molecular Biology and Biophysics, University of Minnesota—Twin Cities, Minneapolis, MN 55455, USA

**Keywords:** sirtuin, PARP, CD38, nicotinamide, nicotinamide riboside, nicotinamide mononucleotide, type 2 diabetes

## Abstract

Obesity-induced insulin resistance is among the key factors in the development of type 2 diabetes, atherogenic dyslipidemia and cardiovascular disease. Adipose tissue plays a key role in the regulation of whole-body metabolism and insulin sensitivity. In obesity, adipose tissue becomes inflamed and dysfunctional, exhibiting a modified biochemical signature and adipokine secretion pattern that promotes insulin resistance in peripheral tissues. An important hallmark of dysfunctional obese adipose tissue is impaired NAD^+^/sirtuin signaling. In this chapter, we summarize the evidence for impairment of the NAD^+^/sirtuin pathway in obesity, not only in white adipose tissue but also in brown adipose tissue and during the process of beiging, together with correlative evidence from human studies. We also describe the role of PARPs and CD38 as important NAD^+^ consumers and discuss findings from experimental studies that investigated potential NAD^+^ boosting strategies and their efficacy in restoring impaired NAD^+^ metabolism in dysfunctional obese adipose tissue. In sum, these studies suggest a critical role of NAD^+^ metabolism in adipose biology and provide a basis for the potential development of strategies to restore metabolic health in obesity.

## 1. Introduction

Obesity is one of the major health burdens worldwide, leading to increased morbidity and mortality [1,2], particularly during the COVID-19 pandemic [3]. Calorie-rich diets, along with a sedentary lifestyle and inadequate intake of health-promoting food bioactives, overwhelm the adaptive mechanisms resulting in cardiometabolic disorders. Obesity-induced insulin resistance has been recognized as one of the major causative factors for the development of type 2 diabetes (T2D), metabolic dysfunction-associated fatty liver disease, atherogenic dyslipidemia and cardiovascular disease [4,5,6]. Adipose tissue undergoes dramatic phenotypic, histological, and metabolic changes in obesity and has an essential role in the regulation of whole-body metabolism. Healthy, non-obese white adipose tissue (WAT) is insulin sensitive and contributes to whole-body insulin sensitivity, whereas the obese, inflamed, and dysfunctional WAT is insulin resistant and leads to the development of T2D and accompanying cardiometabolic disorders [7,8]. However, molecular mechanisms underlying the effects of obesity on metabolic disturbances in adipose tissue, which further influence whole-body metabolism, are still not completely clear. Therefore, studies directed toward understanding the molecular metabolism of adipose tissue in both the lean and obese states have the potential to provide essential information for the development of novel therapeutic strategies for patients with insulin resistance and metabolic syndrome. Among the plethora of regulators of cellular metabolism that are affected in obesity, sirtuins, which had already been identified as cellular energy sensors [9], have emerged as prominent players. Indeed, an important finding within the context of overall adipose tissue dysfunction in obesity is the down-regulation of the NAD^+^ (nicotinamide adenine dinucleotide, oxidized form)/SIRT (sirtuin) system [10]. In this chapter, we focus on the current evidence about the role of NAD^+^ (Figure 1) and NAD^+^-associated molecules in dysfunctional adipose tissue, exploring the potential strategies for restoring metabolic health in obesity. To this end, we conducted a systematic literature search on PubMed using the following search terms: 1. (nad) AND (adipo*); 2. (nad) AND (adipo*) AND (cd38); 3. (nad) AND (cd38) AND (sirt*); 4. (nad) AND (cd38); 5. (parp) AND (adipo*). Following the initial screening, we selected studies summarizing the role of NAD^+^ and NAD^+^-associated molecules in the metabolic regulation of adipose tissue.

## 2. Impairment of the NAD^+^/SIRT Pathway in the Obese Adipose Organ

In mammalian cells, four classes of lysine deacetylases have been described, of which class III consists of seven mammalian sirtuins (SIRT1-7) [11]. Sirtuins are an evolutionarily conserved class of NAD^+^-dependent deacetylases that also act as lysine deacylases and mono-ADP-ribosyltransferases. In particular, SIRT1-3 are strong deacetylases, whereas the others demonstrate a weak or hardly detectable deacetylase activity [12]. The first identified sirtuin was the NAD^+^-dependent histone deacetylase Sir2 (silent information regulator 2) from *Saccharomyces cerevisiae* [13,14], which promotes longevity [15]. In mammalian cells, sirtuins are ubiquitous and manifest their catalytic activity not only on histones, but also on a variety of non-histone proteins. Since their activity is dependent upon NAD^+^, they serve as cellular NAD^+^ sensors and translate their levels to the regulation of diverse cellular processes. Given their broad role in biology, many research groups have reported on their subcellular localization and functions, reviewed in detail elsewhere [12,16,17].

Since sirtuins use NAD^+^ as a co-substrate, they have been identified as NAD^+^ consumers. Within the deacetylation reaction, NAD^+^ is cleaved, producing nicotinamide (NAM), a deacetylated protein and 2′-*O*-acetyl-ADP-ribose [18] (Figure 2). In subsequent reactions, NAM is recycled for NAD^+^ biosynthesis (Figure 3), which is essential for the maintenance of cellular metabolism.

### 2.1. Animal Studies of NAD^+^/SIRT Pathway

In animal studies, there is strong experimental evidence that NAMPT (nicotinamide phosphoribosyltransferase), the key and rate-limiting enzyme in NAD^+^ biosynthesis, is down-regulated in obesity. Yoshino et al. (2011) have demonstrated a decrease of NAMPT protein in WAT of mice with high-fat diet (HFD)-induced T2D, with concomitant decrease of NAD^+^ concentration [19]. In addition, both NAMPT protein and NAD^+^ levels were decreased in the WAT of aged mice, which opened perspectives for studying the NAD^+^ metabolism in the context of age-related metabolic disorders [19]. Consistent with these findings, Song et al. (2014) reported up-regulation of NAMPT at both the mRNA and protein levels in visceral adipose tissue (VAT) of rats following calorie restriction (CR), with a concomitant increase of NAD^+^ concentration [20]. CR was accompanied by a decrease in oxidative stress and an increase in mitochondrial biogenesis in VAT. In addition, Song and coworkers used FK866 to inhibit NAMPT activity, which completely reversed the CR-induced decrease of oxidative stress and reduced the CR-induced increase of mitochondrial biogenesis. At the same time, NAMPT inhibition significantly decreased SIRT1 activity and deacetylation of its targets FOXO1 (forkhead box O1) and PGC1a (PPARg coactivator 1 alpha). Similarly, NAMPT inhibition significantly decreased SIRT3-mediated deacetylation of SOD2 (superoxide dismutase 2), which provides strong evidence for the role of the NAD^+^/SIRT pathway in the regulation of redox status in VAT.

Drew et al. (2016) further explored the NAD^+^/SIRT system in mice fed an HFD and demonstrated significant tissue specificity during both the early and late stages of development of obesity and glucose intolerance [21]. Twenty-five enzymes that are part of the NAD^+^/SIRT system were analyzed at the transcriptional level using a multiplex gene expression assay. In general, a more prominent response was identified in the liver compared to WAT and skeletal muscle. In the WAT, however, HFD modulated PNP (purine nucleoside phosphorylase), NNMT (nicotinamide n-methyltransferase), NMNAT2 (nicotinamide nucleotide adenylyltransferase 2), NMNAT3, SIRT4, SIRT5, and SIRT6. Interestingly, NNMT, the enzyme that catalyzes the reaction of methylation of NAM for its excretion, is up-regulated in the WAT in obesity and T2D. Moreover, NNMT knockdown in mice fed HFD protects against diet-induced obesity, with a concomitant increase of WAT NAD^+^ and modulated expression of several upstream and downstream genes, including SIRT1 targets [22]. In addition, dysregulation of the NAD^+^/SIRT system in obesity has profound effects on different fat depots [23].

To characterize the effect of adipocyte NAMPT on whole-body metabolism, an adipocyte-specific NAMPT deletion model (ANKO mouse) has been established [24]. ANKO mice develop severe multi-organ insulin resistance and adipose tissue inflammation. Plasma concentrations of free fatty acids and triglycerides increase, whereas plasma adiponectin and adipsin decrease. ANKO mice display an increased VAT PPARg (peroxisome proliferator-activated receptor gamma) Ser273 phosphorylation, a post-translation modification that impairs its insulin-sensitizing effects and modulated expression of most of its obesity-linked targets at the mRNA level. At the same time, phosphorylation of cyclin-dependent kinase 5, a critical regulator of PPARg Ser273 phosphorylation, is also increased. Importantly, ANKO mice display a concomitant increase in VAT PPARg acetylation. Since previous studies suggest that PPARg Lys293 hyperacetylation leads to increased Ser273 phosphorylation, it can be hypothesized that NAMPT’s influence on adipocyte and whole-body metabolism is mediated by SIRT1 and possibly other sirtuins. Treatment with rosiglitazone, a well-known PPARg agonist, NMN (nicotinamide mononucleotide) or NA (nicotinic acid), restores the described metabolic disturbances in ANKO mice [25]. Similarly, NR (nicotinamide riboside) improves metabolic flexibility in mice fed a mildly obesogenic diet [26].

### 2.2. In Vitro Studies on 3T3-L1 Adipocytes

To evaluate disturbances of adipocyte metabolism in inflamed obese adipose tissue, the model of tumor necrosis factor alpha (TNFα)-treated 3T3-L1 adipocytes is widely used. Gouranton et al. (2014) demonstrated that TNFα treatment of cultured 3T3-L1 adipocytes leads to a time- and dose-dependent decrease of NAMPT mRNA [27]. In parallel, intracellular NAMPT protein and NAD^+^ concentration, CEBPa (CCAAT enhancer binding protein alpha) mRNA and protein, and SIRT1 activity also decrease, whereas SIRT1 mRNA remains unaffected. At the same time, PTP1B (protein-tyrosine phosphatase 1B), a negative regulator of insulin signaling, is up-regulated at both the mRNA and protein levels, which is mediated by the activity of SIRT1. It has also been demonstrated that NAMPT inhibition via either RNAi or FK866 induces an additional increase of PTP1B expression at both the mRNA and protein levels. TNFα is an important, but not the only, mediator of inflammation in dysfunctional obese adipose tissue [28,29]. Indeed, IL1β (interleukin 1 beta) similarly decreased NAMPT and SIRT3 mRNA expression, as well as significantly increased CD38 mRNA [30]. Extending this observation, TNFα treatment of mature 3T3-L1 adipocytes decreased intracellular NAD^+^ levels and modulated the mRNA expression of genes involved in NAD^+^ metabolism (Figure 4A–C). IL6 (interleukin 6) or IL1β similarly decreased NAMPT and SIRT3 mRNA expression and significantly increased CD38 mRNA (Figure 4D–E), indicating potential adipocyte metabolic and mitochondrial dysfunction [31]. Consistent with this, Zhang et al. (2020) used the model of SIRT3 overexpression in adipocytes and demonstrated that SIRT3 is a positive regulator of autophagy in mature adipocytes. As such, SIRT3 represents a promising target for the treatment of metabolic dysfunction in obesity [32]. Collectively, these data emphasize the role of TNFα, IL1β and IL6 as major modulators of the NAD^+^/SIRT axis and insulin signaling in mouse adipocytes.

### 2.3. Studies in Human Biology

Human studies on visceral adipose tissue metabolism are difficult to conduct, given the difficulty in obtaining samples for analysis. Therefore, most human studies report only results for subcutaneous adipose tissue (SAT). This is a serious shortcoming, given the significant differences in metabolic characteristics of adipose tissue from different fat depots [33]. Despite the limited sample availability, the regulation of NAD^+^ levels has been suggested as a major differentiator between adipose depots. Indeed, weight loss in response to bariatric surgery revealed a significant increase of NMNAT2 mRNA in SAT but not in VAT. At the same time, the expression of genes associated with mitochondrial energy metabolism was also increased, especially in SAT. Therefore, it has been hypothesized that the increase of intracellular NAD^+^ could boost adipose tissue mitochondrial activity [34]. Similarly, NAMPT protein levels are decreased in the visceral adipose tissue of patients with nonalcoholic fatty liver disease linked to the dysfunction of adipose tissue [35].

In 2016, the results of two human studies were published [10,36], the first of which involved young BMI-discordant monozygotic twins [10]. In this study, microarray analysis of SAT revealed decreased mRNA expression of SIRT1, SIRT3, SIRT5, NAMPT, NMNAT2, NMNAT3 and NMRK (nicotinamide riboside kinase) in heavier co-twins compared to their respective leaner co-twin. At the same time, total PARP (poly (ADP-ribose) polymerase) activity tended to be increased in heavier co-twins. In parallel, the expression of several genes that are involved in mitochondrial unfolded protein response (UPRmt) was significantly lower in heavier co-twins. Importantly, UPRmt is a downstream pathway of the NAD^+^/SIRT axis and promotes mitochondrial protein homeostasis against proteotoxic stress that is caused by misfolded proteins [37]. Therefore, reduced UPRmt activity in heavier co-twins suggests impairment of mitochondrial protein quality control in obesity, possibly due to a reduction in the NAD^+^/SIRT pathway activity. In the second study, Rappou et al. (2016) compared clinically healthy obese and lean subjects and monitored the effects of weight loss treatment [36]. Obese subjects had significantly lower NAMPT, SIRT1, SIRT3 and SIRT7 mRNA in SAT, whereas total PARP activity was significantly higher. Consistent with these findings, weight loss treatment caused an increase in SAT SIRT1 and NAMPT mRNAs, whereas total PARP activity decreased. In addition, weight loss was accompanied by a significant decrease in the expression of oxidative stress-related pathways. Importantly, the change in SAT SIRT1 mRNA during the weight loss treatment was positively correlated with the change in brown adipose tissue (BAT) glucose uptake. This finding sheds light on the metabolic function of BAT in adults and the possible involvement of the NAD^+^/SIRT pathway in energy dissipation.

Like white adipose tissue, which has been considered a metabolically inactive depot of triglycerides, the metabolic function of brown adipose tissue in humans has also been incompletely understood. The presence and the role of BAT had been described in newborns, but it was believed that it is completely absent in adults. The main function of brown adipocytes is the uncoupling of the respiratory chain and oxidative phosphorylation, which leads to energy dissipation and the generation of heat. This process is important for the regulation of body temperature in newborns. At the molecular level, the central role in energy dissipation is attributed to UCP1 (uncoupling protein 1), also known as thermogenin, a transmembrane protein located in the inner mitochondrial membrane. UCP1 increases the permeability of the inner mitochondrial membrane for the protons that have been translocated in the mitochondrial intermembrane space, thus reducing the proton motive force, decreasing ATP synthesis, and increasing heat production [38]. With the development of advanced imaging techniques, such as PET/CT scanning, the presence of various amounts of brown adipose tissue has also been detected in adults [39]. It has been demonstrated that a stay in a cold environment increases the levels of BAT within minutes. However, the activity of BAT is lower in men who are overweight or obese [40].

In parallel with the discovery of BAT in adult humans, the phenomenon of “beiging” of WAT has attracted much interest. Beiging of WAT is an adaptive and reversible response to environmental stimuli, such as cold exposure or sympathetic innervation [41]. Interestingly, it has been demonstrated that the gene expression pattern and immunohistochemical characteristics of adult human BAT are more like rodent beige adipose tissue than the classic BAT present in human newborns and rodents [42]. In comparison to the white adipocytes, beige adipocytes have an increased expression of several “brown genes”, which gives them attributes of energy-dissipating cells. Therefore, the beige adipose tissue is less effective in energy storage and has the potential to attenuate obesity and insulin resistance, making it a promising therapeutic target [43].

### 2.4. The Role of NAD^+^ in Regulating Brown Adipose Tissue

While human studies on the NAD^+^/SIRT pathway in VAT are rare, studies of BAT and/or beiging of WAT are similarly difficult. However, several comprehensive experimental studies provide strong evidence of the essential role of NAD^+^ in the metabolic function of BAT. To this end, Yamaguchi et al. (2019) used the previously established ANKO mouse, which lacks NAMPT in both WAT and BAT and demonstrated that ANKO mice have severe cold intolerance and impaired thermogenic responses to fasting and β-adrenergic stimulation [44]. It has also been demonstrated that the BAT of ANKO mice has extremely low NAD^+^ concentration and exhibits clear signs of hypertrophy and whitening. In the same study, Yamaguchi et al. also used a mouse model lacking NAMPT only in the BAT (i.e., BANKO mouse) and demonstrated normal thermogenic responses due to the essential role of NAMPT in catecholamine-induced lipolysis in the WAT, acting as a regulator of NAD^+^/SIRT1/Caveolin-1 axis to provide fatty acids as a fuel supply for BAT thermogenesis. Treatment of ANKO mice with NMN rescued the above-described metabolic abnormalities, thus confirming the indispensable role of NAD^+^ in the regulation of adaptive thermogenesis and whole-body energy metabolism [44]. Similarly, Crisol et al. (2018) have demonstrated that supplementation with NR induces thermogenesis in mice, which is accompanied by an increase in UCP1 protein and PGC1a mRNA in the BAT [45].

In addition to enzymes whose roles in NAD^+^ metabolism have been well defined, Nguyen et al. (2020) have characterized AIFM2 (apoptosis-inducing factor mitochondria associated 2) and its potential role as a regulator of NAD^+^ levels [46]. AIFM2 is highly expressed in mouse BAT as well as mouse inguinal WAT (iWAT) and human SAT. Upon cold exposure or β-adrenergic stimulation, AIFM2 translocates from the lipid droplets to the outer side of the mitochondrial inner membrane, where it acts to regenerate the cytosolic NAD^+^ and provide electrons to the electron transport chain, thus supporting the glycolysis that is needed for thermogenesis. Interestingly, although in the fasted state, BAT uses primarily the free fatty acids released from WAT for thermogenesis, Nguyen et al. propose that in the fed state with high circulating glucose, the glucose may serve as a major fuel for thermogenesis, with AIFM2 playing a critical role by supporting glycolysis. Indeed, AIFM2-knockout mice have impaired thermogenesis, alongside decreased BAT and iWAT NAD^+^/NADH, increased body weight and adiposity [46]. Therefore, understanding the role of AIFM2 in the adipose tissue may provide future therapeutic targets for diet-induced obesity and insulin resistance.

Wang et al. (2013) demonstrated that erythropoietin (EPO) has beneficial effects in obesity-induced insulin resistance in mice, and these effects are NAD^+^/SIRT1-mediated [47]. Erythropoietin receptor (EPOR) is highly expressed in the mouse WAT, with approximately 60% of its expression in hematopoietic tissue [48], suggesting that, in addition to its role in erythropoiesis, EPO also plays an important metabolic role. A mouse model of adipocyte-specific deletion of EPOR exhibits obesity, glucose intolerance and insulin resistance, especially when mice are challenged with a high-fat diet. Concordantly, treatment with EPO improves metabolic parameters in diet-induced obese wild-type mice. Aiming to reveal molecular mechanisms underlying the beneficial effects of EPO on metabolic health, Wang et al. demonstrated that EPO had virtually no effect on the BAT but significantly regulated SAT suggesting this depot as the main EPO target. Moreover, EPO induced an increase in the intracellular and tissue NAD^+^ needed for the activation of SIRT1 and the acquisition of brown adipocyte gene expression in white adipocytes.

## 3. The Role of Adipose PARPs and CD38 as NAD^+^ Consumers

The maintenance of intracellular NAD^+^ concentrations is critical for the proper activity of sirtuins, which have the capacity to ameliorate the negative effects of overnutrition and physical inactivity on mitochondrial function and metabolic health. Intracellular NAD^+^ is regulated by both NAD^+^ biosynthesis and NAD^+^ consumption, and in this section, we focus on two major NAD^+^ consumers, PARPs and CD38.

### 3.1. PARPs in Adipose Tissue

PARPs catalyze mono and poly-ADP-ribosylation, a post-translational protein modification where ADP-ribose is transferred to the target protein in a reaction that consumes NAD^+^ to produce NAM and ADP-ribose [49]. In addition to classical PARP functions that primarily include DNA repair, transcriptional regulation and apoptosis, recent studies point to a role in inflammation, mitochondrial function, cancer biology, circadian rhythm and aging. Focusing on the metabolic properties of PARPs, the interaction between PARP1 and SIRT1 emerges as an important regulatory mechanism. Using the same co-substrate, the balance between these two metabolic regulators is primarily determined by their K_m_ values for NAD^+^. Interestingly, the PARP1 K_m_ for NAD^+^ is lower than that for SIRT1, and upon its activation, PARP1 rapidly consumes the intracellular NAD^+^, thus disabling the SIRT1 activity. Conversely, SIRT1 can act as a negative regulator of PARP1 through the inhibition of transcription or deacetylation. Together, PARP1 and SIRT1 reciprocally regulate NF-κB, the pivotal mediator of inflammatory response, with PARP1 participating as a transcriptional co-activator of NF-κB, while SIRT1 inhibits NF-κB activity by deacetylating RelA/p65 [50].

The role of PARP1 in the metabolic function of adipose tissue has been studied in the mouse model of PARP1 deletion [51]. PARP1^−/−^ mice are leaner and exhibit increased energy expenditure compared to their PARP1^+/+^ littermates. The analysis of BAT shows markedly increased NAD^+^ levels, increased SIRT1 activity (as demonstrated by reduced acetylation of its targets PGC1a and FOXO1) and increased mitochondrial biogenesis. At the transcriptional level, there is significant modulation of the genes involved in mitochondrial respiration, uncoupling, fatty acid oxidation and thyroid hormone activation. The metabolic involvement of PARP1 has also been demonstrated in the offspring of mice fed a high-calorie diet prior to and during gestation and lactation [52]. PARP1 protein level was increased in the SAT of offspring of mice fed a high-calorie diet, which was accompanied by complex metabolic disturbances, including decreased NAD^+^/NADH ratio, increased acetylation of FOXO1 indicative for decreased SIRT1 activity and impaired mitochondrial bioenergetics. Supplementation of the maternal diet with niacin partially reversed the metabolic disturbances resulting from the high-calorie diet, decreasing the PARP1 protein level and increasing the NAD^+^/NADH ratio and SIRT1 activity.

### 3.2. CD38 in Adipose Tissue

Besides its role as a co-substrate of sirtuins and PARPs, NAD^+^ also serves as a precursor for the biosynthesis of cyclic ADP-ribose (cADP-ribose) [53], a molecule that plays an important role in calcium signaling as a second messenger. The enzyme that catalyzes the reaction of biosynthesis of cADP-ribose was first discovered in the marine animal *Aplysia* and was named ADP-ribosyl cyclase [54]. This enzyme is a soluble protein, and therefore, the finding that CD38 is its mammalian homolog was rather surprising as CD38 is a transmembrane protein that was originally identified as a lymphocyte surface antigen. Further experiments have demonstrated the extraordinary multi-functionality of CD38. Indeed, CD38 has the ability to catalyze the conversion of NAD^+^ to both cADP-ribose (referred to as cyclase activity) and ADP-ribose (referred to as NADase activity) [55]. It has also been demonstrated that CD38 can hydrolyze the cADP-ribose to ADP-ribose (Figure 5). Consistent with CD38 as a major regulator of NAD^+^, Aksoy et al. (2006) demonstrated an increase in tissue NAD^+^ levels in CD38 knockout mice relative to wild-type animals [56].

CD38 has been extensively studied in the context of research, diagnostics and treatment of multiple myeloma [57], chronic lymphocytic leukemia [58], acute myeloid leukemia [59], autism spectrum disorder [60] and allergic airway disease [61]. However, there are only a handful of papers about the involvement of CD38 in metabolic disturbances in obesity. The San Antonio Family Heart Study was the first study that linked blood lipids (triglycerides and HDL) with CD38 and the risk for the development of metabolic syndrome [62]. Of note is the pioneering work of Barbosa et al. (2007), who demonstrated that CD38 knockout mice are resistant to high-fat diet-induced obesity, as they exhibit an enhanced energy expenditure. This phenomenon is mediated, at least in part, by the NAD^+^-dependent SIRT1/PGC1a axis that regulates mitochondrial biogenesis and energy homeostasis [63]. A follow-up study confirmed that CD38 knockout mice are protected from diet-induced obesity and demonstrated high NAD^+^ levels in their WAT and BAT on a high-fat, high-sucrose diet [64].

CD38 is among the genes that are significantly up-regulated in adipose tissue of high-fat diet-induced obese rats [65]. Consistent with this finding, high-fat-fed obese mice exhibit a 2-fold increase in CD38 mRNA in WAT [66]. Slightly discrepant results were reported by Drew et al. (2016), who demonstrated only a minor increase in CD38 mRNA in WAT [21]. CD38 mRNA in adipose tissue is up-regulated not only in animal models of diet-induced obesity but also in some genetic models. The publicly available database from the University of Wisconsin (http://diabetes.wisc.edu, accessed on 8 October 2022) provides information that CD38 mRNA is increased in adipose tissue of genetically obese mice of B6 strain in comparison to lean mice but not in genetically obese BTBR mice. In vitro experiments using cultured 3T3-L1 adipocytes demonstrated a marked increase in CD38 mRNA upon TNFα, IL6 or IL1β treatment [30,31].

Even though not analyzed in the context of diet-induced obesity, Benzi et al. (2021) demonstrated a marked down-regulation of CD38 at both mRNA and protein levels in the BAT of wild-type mice during cold exposure, accompanied by an increase in NAD^+^ levels. The effect of cold exposure was slightly different in WAT, where the significant down-regulation of CD38 was accompanied by a strong increase in NAD(P)H levels. Experiments conducted using CD38 knockout mice supported these findings, leading to the conclusion that CD38 can be considered a relevant molecular target to combat obesity-induced insulin resistance [67]. An overview of rodent studies on the role of NAD^+^ in adipose tissue is presented in Table 1.

## 4. Strategies for NAD^+^ Boosting in Adipose Tissue

Several research groups have directed their interest toward exploring NAD^+^-boosting strategies for regulating adipose energy metabolism. Calorie restriction [20] has been reported as a potential NAD^+^ boosting strategy but, in general, it is difficult to implement in humans. Even though some of the earlier studies have analyzed the metabolic effects of different NAD^+^ precursors on adipose tissue [25,26,44,45,52], latter studies of the role of NAD^+^ in the metabolic regulation of dysfunctional obese adipose tissue have primarily focused on the effects of supplementation with NAD^+^ precursors. For example, studying the metabolic effects of NAM supplementation in B57BL/6J male mice fed a high-fat diet, Méndez-Lara et al. (2021) demonstrated its capacity to reduce body weight gain and thus protect against diet-induced obesity. The mechanisms involved, among others, include increased global energy expenditure and induction of iWAT beiging, as well as increased mitochondrial β-oxidation of fatty acids and increased NAD^+^ levels in the iWAT [68]. In a similar experimental design, Luo et al. (2022) confirmed that NAM supplementation can ameliorate diet-induced obesity in B57BL/6J male mice. They also confirmed previously observed effects of NAM supplementation in adipose tissue, such as enhanced mitochondrial function and increased NAD^+^ levels. Surprisingly, this study also revealed that NAM supplementation can increase glutathione biosynthesis in the adipose tissue, which is of importance for maintaining cellular redox homeostasis [69].

Even though animal studies have clearly demonstrated the beneficial metabolic effects of NAD^+^ precursors on dysfunctional obese adipose tissue, translation of this experimental evidence to the human population has proven challenging [70,71]. Within these efforts, it will be essential to take into consideration the safety, efficacy and side effects of supplementation with NAD^+^ precursors in humans [71], different delivery routes [72], but also factors that determine interindividual variabilities [73,74], such as sex, age, comorbidities, medication and epigenetic factors.

## 5. Summary and Conclusions

NAD^+^ is one of the key non-protein molecules in cellular biology, which, since its discovery, has continuously surprised with new and previously unknown functions. NAD^+^ acts as a cofactor for several oxidoreductases that catalyze key biochemical reactions of the cellular energy metabolism and transports the reducing equivalents to the electron transport chain, thus being an important mediator in ATP biosynthesis. In addition, several non-redox functions of NAD^+^ have been reported, including being a donor of ADP-ribose in the reactions of ADP-ribosylation, a precursor of the second messenger cyclic ADP-ribose, and a co-substrate of sirtuins. The discovery of sirtuins and the elucidation of their versatile cellular regulatory functions were crucial for increased interest in this already well known “old” molecule.

Compared to other metabolically active tissues, the non-redox functions of NAD^+^ in adipose tissue remain the least investigated and understood, mainly due to the inherent nature of adipose tissue, characterized by its remarkable dynamics and heterogeneity. Moreover, there is a significant involvement of NAD^+^ in the metabolic remodeling of adipose tissue, whether it is in the context of obesity-induced dysfunction or the process of beiging of white adipose tissue. Further investigation into the non-redox functions of NAD^+^ in adipose tissue represents one of the primary goals in adipose biology, particularly given the development of strategies for NAD^+^ boosting in the human population (Figure 6). The potential for the development of strategies for reversal of impaired NAD^+^ metabolism in obese adipose tissue and consequent improvement of metabolic health in humans seems achievable.

## Figures and Tables

**Figure 1 biomedicines-11-02560-f001:**
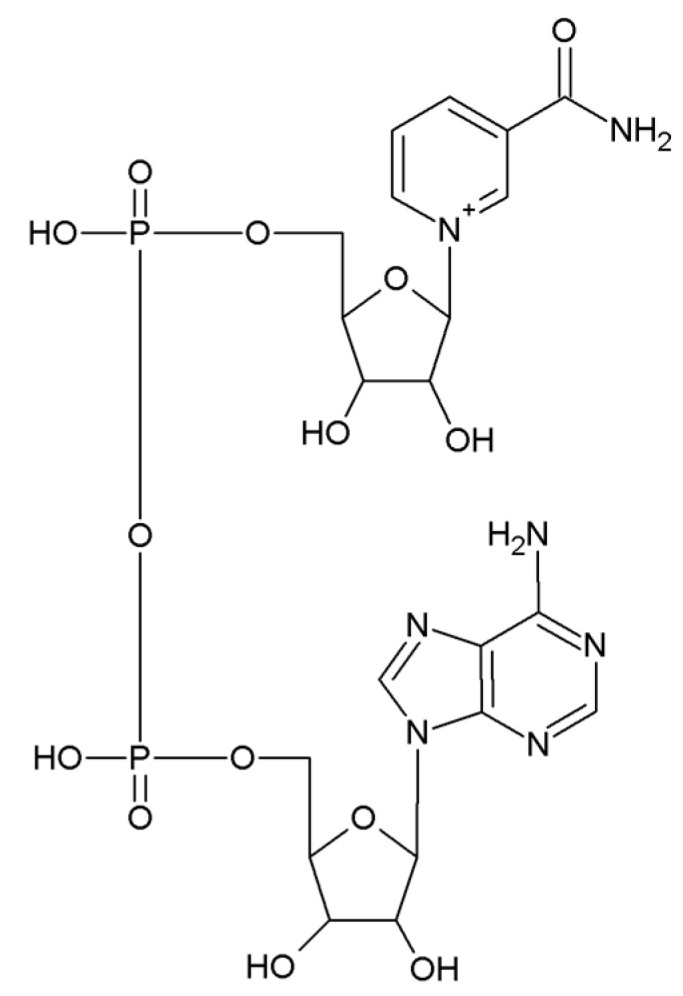
Chemical structure of NAD^+^.

**Figure 2 biomedicines-11-02560-f002:**
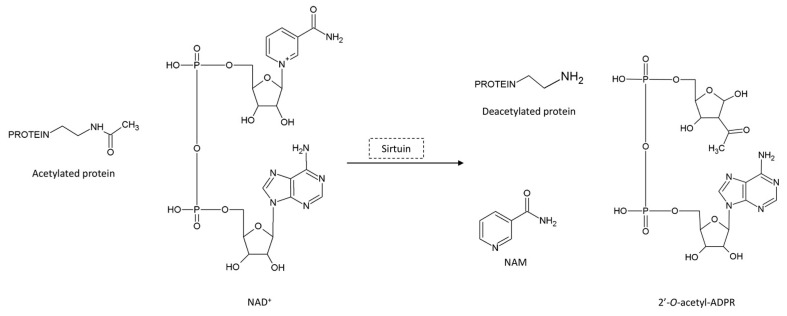
Sirtuin-mediated protein deacetylation.

**Figure 3 biomedicines-11-02560-f003:**
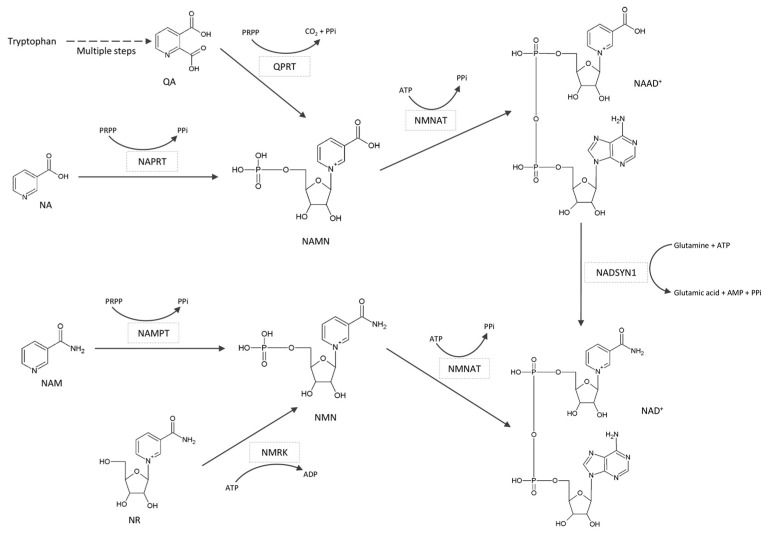
Pathways for NAD^+^ biosynthesis in mammalian cells.

**Figure 4 biomedicines-11-02560-f004:**
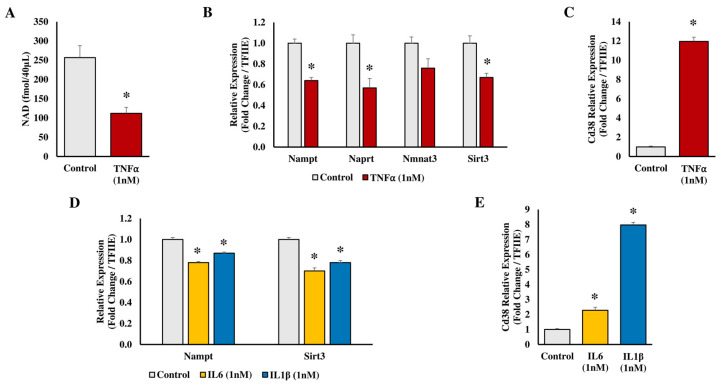
Effects of TNFα, IL6 and IL1β on 3T3-L1 adipocytes in culture. Differentiated 3T3-L1 adipocytes were treated with the indicated levels of inflammatory factors for 24 h, and extracts prepared for metabolite and mRNA qPCR analysis. (**A**) NAD^+^ in TNFα-treated 3T3-L1 adipocytes, (**B**) Nampt, Naprt, Nmnat3, and Sirt3 in TNFα-treated 3T3-L1 adipocytes, (**C**) Cd38 in TNFα-treated 3T3-L1 adipocytes, (**D**) Nampt and Sirt3 in IL6- or IL1β-treated 3T3-L1 adipocytes, (**E**) Cd38 in IL6- or IL1β-treated 3T3-L1 adipocytes. * *p* < 0.01.

**Figure 5 biomedicines-11-02560-f005:**
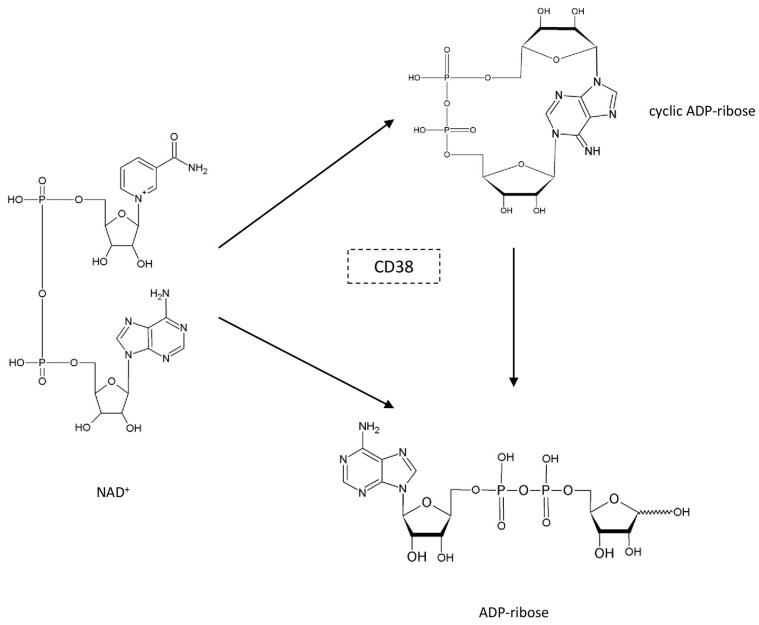
Multi-functionality of the CD38-catalyzed reaction.

**Figure 6 biomedicines-11-02560-f006:**
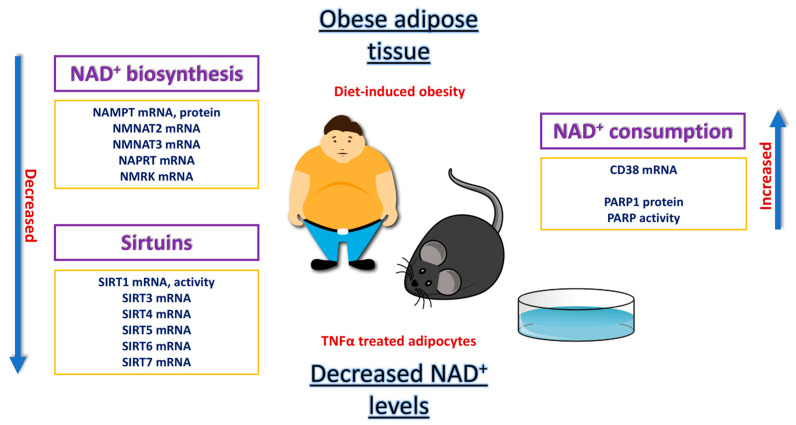
Summary of the evidence for impaired NAD^+^ metabolism in dysfunctional adipose tissue.

**Table 1 biomedicines-11-02560-t001:** Overview of rodent studies on the role of NAD^+^ in adipose tissue.

Model	Type of Adipose Tissue	Major Metabolic Outcome/s
High-fat diet (HFD)	White adipose tissue (WAT)	Type 2 diabetes Decreased NAD^+^ levels [19] Obesity Glucose intolerance Altered regulation of the SIRT/NAD^+^ system [21]
Calorie restriction (CR)	Visceral adipose tissue (VAT)	Increased NAD^+^ Decreased oxidative stress Increased mitogenesis [20]
Nnmt knockdown and high-fat diet	White adipose tissue (WAT)	Reduced diet-induced obesity Enhanced insulin sensitivity Increased energy expenditure Increased NAD^+^ [22]
Adipocyte-specific Nampt knockout (ANKO)	Visceral adipose tissue (VAT) Subcutaneous adipose tissue (SubQ)	Multi-organ insulin resistance Adipose tissue inflammation Decrease of plasma adiponectin and adipsin [25]
Adipocyte-specific Nampt knockout (ANKO)	Brown adipose tissue (BAT)	Cold intolerance Impaired thermogenic response Decreased NAD^+^ in BAT Hypertrophy and whitening of BAT [44]
High-fat diet supplemented with nicotinamide riboside (NR)	Epididymal white adipose tissue (eWAT)	Improved metabolic flexibility with NR addition [26]
Regular diet supplement with nicotinamide riboside (NR)	Brown adipose tissue (BAT)	Reduced abdominal visceral fat Increased heat production [45]
Whole body Aifm2-knockout	Brown adipose tissue (BAT) Inguinal white adipose tissue (iWAT)	Impaired thermogenesis Decreased BAT and iWAT NAD^+^/NADH ratio Increased body weight and adiposity [46]
Aifm2-overexpression in UCP1^+^ cells	Brown adipose tissue (BAT) Inguinal white adipose tissue (iWAT)	Increased thermogenesis Increased BAT and iWAT NAD^+^/NADH ratio Decreased body weight and adiposity [46]
Adipocyte-specific deletion of erythropoietin receptor	Subcutaneous WAT (SubQ)	Obesity Glucose intolerance Insulin resistance Decreased oxygen consumption Decreased NAD^+^ in SubQ [47]
PARP1 knockout mouse	Brown adipose tissue (BAT)	Reduced fat accumulation Higher energy expenditure Higher NAD^+^ in BAT Higher mitochondrial content in BAT [51]
Maternal high-calorie diet	Subcutaneous WAT (SubQ) of newborn mice at the end of lactation	Increased SubQ mass Increased adipocyte size Decreased NAD^+^/NADH ratio Increased oxidative damage to mitochondrial proteins Impaired mitochondrial bioenergetics [52]
Maternal high-calorie diet supplemented with niacin	Subcutaneous WAT (SubQ) of newborn mice at the end of lactation	Increased NAD^+^/NADH ratio in SubQ [52]
CD38 knockout mice on high-fat high-sucrose diet	White adipose tissue (WAT) Brown adipose tissue (BAT)	Higher levels of NAD^+^ in WAT and BAT Improved metabolic flexibility [64]
CD38 knockout mice on high-fat diet	White adipose tissue (WAT)	Less body weight gain [66]
Cold exposure	Interscapular brown adipose tissue (iBAT)	Down-regulation of CD38 at both mRNA and protein levels in iBAT Increased NAD^+^ in iBAT [67]
Cold exposure, CD38 knockout mice	Interscapular brown adipose tissue (iBAT)	Increased NAD^+^ in iBAT [67]

## Data Availability

Not applicable.

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
