# Peer review of "The Role of NAD+ in Metabolic Regulation of Adipose Tissue: Implications for Obesity-Induced Insulin Resistance"

_biomedicines, 2023, doi:10.3390/biomedicines11092560_

Round 1

Reviewer 1 Report

The manuscript does not have a discussion section. This is the most important part for readers understanding. The authors need to add a discussion section.

Please show the chemical structure of NAD+ in the Introduction Section.

The bibliography majorly has references before 2010. The authors should try to cite more relevant and recent literature in the field. such as  https://doi.org/10.3390/polym15092031 and https://doi.org/10.1016/j.ijbiomac.2022.01.134

-Overall, the expression of sentences and choice of words used in this article must be corrected into more explicit and appropriate, enabling readers to obtain correct information and better understand what the authors are trying to deliver. 

- Prepare a Table and compare the results of recent relevant results.

Minor editing of English language required.

Author Response

Reviewer 1

Comments and Suggestions for Authors

- The manuscript does not have a discussion section. This is the most important part for readers understanding. The authors need to add a discussion section.

We thank the Reviewer for this suggestion. We have now included in the revised manuscript a new section titled "Summary and conclusion”.

- Please show the chemical structure of NAD+ in the Introduction Section.

We thank the Reviewer for this suggestion. We have now included in the Introduction a new figure (now Figure 1) representing the chemical structure of NAD+.

- The bibliography majorly has references before 2010. The authors should try to cite more relevant and recent literature in the field. such as https://doi.org/10.3390/polym15092031 and https://doi.org/10.1016/j.ijbiomac.2022.01.134

In preparing this review article, our aim was to provide a comprehensive presentation of the topic, including a small number of seminal papers such as:

  • Klar, A.J.; Fogel, S.; Macleod, K. MAR1-a Regulator of the HMa and HMalpha Loci in SACCHAROMYCES CEREVISIAE. Genetics 1979, 93, 37-50.
  • Nedergaard, J.; Bengtsson, T.; Cannon, B. Unexpected evidence for active brown adipose tissue in adult humans. Am J Physiol Endocrinol Metab 2007, 293, E444-452, doi:10.1152/ajpendo.00691.2006.

The “rediscovery” of NAD+ through its roles in non-redox reactions opened new perspectives not only in adipose biology, but more importantly in human medicine, and therefore we thank the Reviewer for suggesting this relevant reference (DOI: 10.3390/polym15092031), which is now included in the manuscript.

-Overall, the expression of sentences and choice of words used in this article must be corrected into more explicit and appropriate, enabling readers to obtain correct information and better understand what the authors are trying to deliver.

We regret to hear that the Reviewer finds the expression of sentences and choice of words not appropriate.  We have edited the chapter and made many changes in sentence structure and style.  We hope that the reviewer finds the revised text better presented. 

- Prepare a Table and compare the results of recent relevant results.

We thank the Reviewer for this suggestion. In the manuscript we have now included a table (Table 1) presenting an overview of animal studies, which comprise the majority in this review. The table includes information about the paper (reference), species, animal model, type of adipose tissue studied, and the major metabolic outcomes of the study.

Reviewer 2 Report

I was presented with an exciting article for review. This article, although the authors put a lot of work into it, needs to be refined.
There is no information on the selection of literature. I have a question, what databases were analyzed, and what keywords were used?
What was the authors' aim for which they undertook the presented subject matter?

The figures are illegible. Since it is a review article, tables, diagrams, and summaries would be valuable, but they are missing.
Everything is described in one text, and it's hard to get through.
There is no summary, no conclusions, and no authors' suggestions.
I did not find any research plots outlined in the article.

It is also inappropriate to refer to literature in the text by the authors. They use names, not numbers.

Minor editing of English language is required.

Author Response

Comments and Suggestions for Authors

I was presented with an exciting article for review. This article, although the authors put a lot of work into it, needs to be refined.

We thank the Reviewer for their positive comment. We are glad to address their pertinent suggestions for improving the manuscript.

There is no information on the selection of literature. I have a question, what databases were analyzed, and what keywords were used?

What was the authors' aim for which they undertook the presented subject matter?

We thank the Reviewer for these suggestions. In the final few lines of the Introduction, we now clearly explain the aim of this review and present the search terms that we used for conducting the literature search on PubMed.

The figures are illegible. Since it is a review article, tables, diagrams, and summaries would be valuable, but they are missing.

We thank the Reviewer for this suggestion. In the manuscript, we have now included a table (Table 1) presenting an overview of animal studies on the role of NAD+ in adipose tissue, and a summary figure (Figure 6) of the evidence for impaired NAD+ metabolism in dysfunctional adipose tissue.

Regarding the resolution of figures, it was lost with their conversion to pdf format during the submission process.  In the manuscript, we have now included larger images. For publication, we can provide original figures with good resolution.

Everything is described in one text, and it's hard to get through.

Thank you for this comment. In the revised manuscript we use subheadings to clearly distinguish the subsections.

There is no summary, no conclusions, and no authors' suggestions.

We thank the Reviewer for this suggestion. We have now included in the revised manuscript a new section titled "Summary and conclusion”.

I did not find any research plots outlined in the article.

We thank the Reviewer for this comment. In addition to RNAseq results (ref [30]: Hertzel, A.V.; Yong, J.; Chen, X.; Bernlohr, D.A. Immune Modulation of Adipocyte Mitochondrial Metabolism. Endocrinology 2022, 163, doi:10.1210/endocr/bqac094.), we now include in the manuscript our additional data depicted in Figure 4.

It is also inappropriate to refer to literature in the text by the authors. They use names, not numbers.

This issue has been resolved, and the references are now cited in MDPI style.

Round 2

Reviewer 2 Report

The article was very well suplemented and all my suggestions were taken into account. Everything is correct in the current version, so I recommend accepting the article as it stands.